# PLATO: Predicting Latent Affordances Through Object-Centric Play

**Suneel Belkhale, Dorsa Sadigh**
Stanford University

**Abstract:** Constructing a diverse repertoire of manipulation skills in a scalable fashion remains an unsolved challenge in robotics. One way to address this challenge is with unstructured human play, where humans operate freely in an environment to reach unspecified goals. Play is a simple and cheap method for collecting diverse user demonstrations with broad state and goal coverage over an environment. Due to this diverse coverage, existing approaches for learning from play are more robust to online policy deviations from the offline data distribution. However, these methods often struggle to learn under scene variation and on challenging manipulation primitives, due in part to improperly associating complex behaviors to the scene changes they induce. Our insight is that an object-centric view of play data can help link human behaviors and the resulting changes in the environment, and thus improve multi-task policy learning. In this work, we construct a latent space to model object *affordances* – properties of an object that define its uses – in the environment, and then learn a policy to achieve the desired affordances. By modeling and predicting the desired affordance across variable horizon tasks, our method, Predicting Latent Affordances Through Object-Centric Play (PLATO), outperforms existing methods on complex manipulation tasks in both 2D and 3D object manipulation simulation and real world environments for diverse types of interactions. Videos can be found on our website.

**Keywords:** Human Play Data, Object Affordance Learning, Imitation Learning

## 1 Introduction

The field of robotics has seen tremendous progress in solving manipulation tasks, but learning a general multi-task policy remains an open challenge. Imitation learning methods are sample-efficient at replicating demonstrated behaviors, but are often presented with structured, predefined tasks and therefore struggle to generalize outside of the data distribution [1, 2]. Rather than using predefined task demonstrations, recent work has shown that learning from *play* data – an unstructured form of demonstration without predefined goals – can lead to policies that are more robust to online deviations [3]. Play data is easy to collect at scale since it requires no task specification or manual resetting, and play can have broad data coverage over the set of object interactions necessary for performing a variety of tasks.

Existing approaches for learning from play sample short horizon-length windows from play data to learn goal-conditioned imitation policies in an offline fashion [3, 4]. However, not all facets of the demonstrator's behavior are captured by the goal alone; thus prior work learns a latent space to model the variation in human behaviors. Such a latent space captures a representation of "plans," for example plans representing reaching or grasping motions, for a given goal state during play. These latent plans can then help inform robot policies at test time [3].

These approaches make several restrictive assumptions. Firstly, they assume that given a fixed short-horizon window of play, the agent's goal is the environment state at the end of the window. This assumption however is not always true: since the desired robot state is not available at test time, the goal is chosen as the environment state a few seconds in the future. Since the goal might be close or the same as the initial state, it can be uninformative for the policy, nor does it necessarily represent the human's true goal. Crediting behavior to an incorrect environment goal can obscure *why* a human chose the actions during that window, and thus hinder policy learning. For example,

6th Conference on Robot Learning (CoRL 2022), Auckland, New Zealand.

if we want to grasp and move a block on a table but our goal is sampled after the robot has initiated motion but before initiating the grasp, the sampled goal will have no change in the object state and thus gives us no information on the true goal that the user had in mind. Secondly, by randomly sampling windows from play data, the learned latent space will be forced to model all sequences of behaviors *equally*, even though many sequences will be less critical to achieving the desired environment goal. These restrictive assumptions leads to suboptimal behaviors when increasing the complexity and variability of tasks, e.g., tasks with varying horizons, when learning from play.

Instead of defining goals and plans based on arbitrary horizon lengths, we posit that humans often define goals and plan in terms of interactions with objects: rather than planning over the individual joint motions required to grasp and open a door handle, we think about turning the handle and then opening the door. Our key insight is that viewing offline play data as diverse object interactions enables better modeling of both a human's goals and the behaviors that can achieve these goals. Rather than learning to represent fixed short-horizon robot trajectories, or *plans*, we bias the latent space towards learning demonstrated *object affordances* that accomplish tasks in object-space. Imitation policies can then condition on these affordances directly to gain insight into human's behaviors.

We propose an algorithm, PLATO – Predicting Latent Affordances Through Object-Centric Play – that automatically segments play into a series of object interactions using proprioceptive information, and then learns a latent affordance space over object interactions. Simultaneously, it learns to imitate human actions over variable horizon interactions, conditioned on the latent affordance and goal. By considering object interactions in play and correctly attributing goals to robot behaviors, PLATO builds a robust mapping between the true goals, desired object affordances, and actions on the robot over varying horizons. This leads to PLATO significantly outperforming prior methods especially when increasing the complexity and variability of play data. Our contributions are:

1. We have developed an object-centric paradigm for learning from unstructured human play data, which views play as sequential, unlabeled interactions with objects.
2. We have developed a new algorithm, PLATO, that extracts and leverages these interactions to model diverse object affordances from play data and learn a robust policy.
3. We have tested PLATO on a number of 2D and 3D manipulation environments in simulation and the real world, including diverse objects such as blocks, mugs, cabinets, and drawers, with broad coverage over possible tasks and object affordances. We demonstrate that PLATO substantially outperforms state-of-the-art learning from play baselines in these complex manipulation tasks.

## 2 Related Work

In this section, we will discuss prior work in goal-conditioned imitation learning, learning from play, and object-centric policy learning. Our work brings an object-centric perspective on learning from play to learn effective imitation policies that generalize across different manipulation tasks.

**Imitation Learning.** Imitation Learning is a common method for learning robot policies from human demonstrations, where a policy learns to mimic human actions [1]. These methods often struggle to generalize to new environments and to learn from multi-task data [5, 6, 2]. To improve *generalization*, Ho, et al. introduced generative adversarial IL to learn from imitation data while matching the expert policy distribution [7]. Recently, implicit imitation learning policies using energy-based models have also been shown to improve generalization as compared to explicit policy models [8]. To enable *multi-task* learning, one can condition the policy on goal states, either explicitly labelled or inferred via hindsight experience replay [9, 10, 11]. Other methods have leveraged meta-learning for multi-task imitation learning to enable better multi-task performance and one-shot generalization [12]. A key limitation of these works is the dependency on demonstration quality and quantity, for example the state-action coverage within the dataset for each task [13]. Our work utilizes an approach built on top of goal-conditioned imitation learning to learn *multi-task* policies. However, we use play data as opposed to expert demonstrations, which enables a much broader state-action coverage to learn *generalizable* and robust policies.

**Learning from Play.** Human play data is defined as unstructured and unsupervised human interaction with an environment, as a means to let the user guide the data collection process, and provide broader coverage of the task-relevant state and goal space. Because humans can freely choose how to interact with the environment, play data is easy and cheap to collect without the need for prior task specification or environment resets. These factors make play data a rich source for skill learning; as

a result, imitation policies trained on play data are more robust to deviations from the expert trajectories than those trained on single-task demonstrations [3]. The state of the art method, Play-LMP, learns latent "plans" over short, fixed horizon trajectories from play data [3]. This approach can also be used to ground language with play data using pretrained language models [14]. These learned short-horizon skills have also been chained together to accomplish long horizon tasks using motion planning inspired techniques [15]. Additionally, using small variations in the horizon length of play windows has been shown to improve test time performance [4]. These methods benefit from the broad coverage of play data, but suffer from distribution shift at test time due to incorrectly inferred goals during training. Therefore, our approach also imitates play, but we take an object-centric view: by learning from diverse interactions with the environment, we can more accurately infer the human's goals during play and thus obtain a better state-action-goal distribution.

**Object-Centric Modeling and Policy Learning.** There is strong evidence that humans have neural pathways specific to object recognition, dynamics understanding, and scene segmentation [16]. Inspired by humans, this notion of object centrism has been applied in many recent works in object manipulation for better policy generalization and sample efficiency. Formalisms like object-oriented MDPs have been introduced to better model *effects* of agent behaviors as functions of individual objects and their affordances [17]. Object affordances have also been learned from labeled interaction examples in multi-task environments and were shown to benefit policy learning downstream [18, 19]. Planning relative to object reference frames and primitives can enable plans to generalize to changing environments [20, 21]. Similarly, focusing on objects during policy learning can allow behaviors to generalize to novel scenarios [22, 23]. A recent work leveraged labelled object motions in play data to predict grasp points, and showed RL becomes more sample efficient post-grasping [24]. We employ object centrism to guide learning from play data, specifically by extracting affordances through unsupervised temporal segmentation over object interactions.

## 3 Predicting Latent Affordances Through Object-Centric Play

A play dataset $D_{\text{play}}$ consists of $N$ varying length episodes of undirected, human generated state-action trajectories $T_j$, $\forall j \in \{1, \ldots, N\}$, where $T_j = \{s_1, a_1, \ldots, s_{L_j}\}$ for length $L_j$. The state feature space $S$ may be learned or predefined, and consists of a proprioceptive state space $S^r$ (robot state space) and an environmental state space $S^o$ (object state space) such that $S^r \oplus S^o = S$. The goals that generated these trajectories are not included in the dataset, and are defined as $o_g \in G$, for goal space $G \subseteq S^o$. We use $o$ and $s^o$ interchangeably to refer to environmental state, and likewise for $r$ and $s^r$ to refer to the robot state. We assume that goals only consist of the environmental state, since access to proprioceptive state goals at test time is unrealistic as it requires the robot having access to a policy for achieving the given environmental goal.

Given access to this play dataset $D_{\text{play}}$, a new initial state $s \in S$, and an object goal state $o_g$, our problem is to learn a robot policy $\pi$ to achieve the desired goal state. In prior work, sampled trajectories of play $\tau$ consist of a contiguous fixed-horizon segment from an episode $T_j$. For simplicity we denote sampling these segments from play as $\tau \sim D_{\text{play}}$, where the length of the segment is the fixed horizon $H$. We propose viewing play sequences from a bird's eye view, namely object interactions, instead of from myopic and fixed 1-2 second windows used in prior work. From this perspective, we hypothesize human play is just a series of *environment interactions* induced by a robot's actions. If we can somehow detect where interactions begin and end, we can learn to relate interaction and pre-interaction robot behaviors to the state of the world post-interaction. First, we discuss how we detect interactions in the environment in offline play by leveraging proprioceptive cues. Next, we formalize how we properly choose and associate goals with robot behaviors.

**Segmenting Play into Interaction Phases**: When we interact with an environment, we change its state through our own actions, whether it be through direct or indirect contact. In this work, we focus on single-object interactions, but we emphasize that interactions can be defined even over multiple objects that are being influenced by the robot behavior (e.g., in tool-mediated manipulation, interaction is defined between the tool and the environment). We break down an interaction into the following phases:

1. **Pre-interaction:** This phase usually involves orienting the robot to interact with an object, e.g., reaching the purple block in Fig. 1.

2. **Interaction:** This phase involves joint and interdependent motions between the robot and the object(s), e.g., pulling the purple block in Fig. 1.

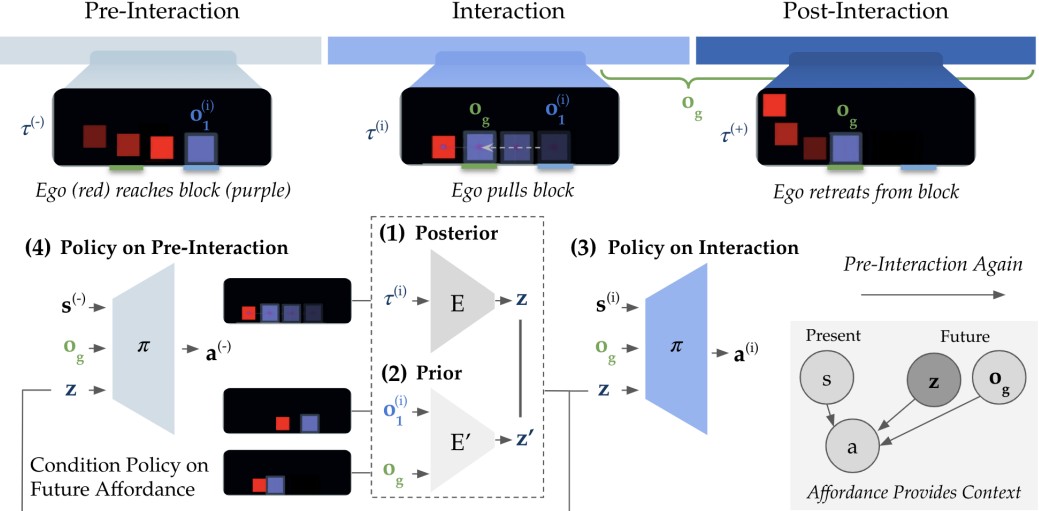

Figure 1: Our method shown on pre-interaction (purple), interaction (blue), and post-interaction (green) periods. (1) Posterior $E$ encodes the interaction sequence $\tau^{(i)}$ into affordance $z$. (2) Prior $E'$ encodes object start and goal states $o_1^{(i)}$ and $o_g$ to predict $z$. $o_g$ is sampled from post-interaction. (3) Policy trained to output actions on interaction period conditioned on affordance, and simultaneously (4) to output actions from pre-interaction period conditioned on the "future" affordance. The assumed causal structure is shown in the lower right: PLATO claims the robot behavior can be explained with knowledge of the long term goal and future affordance $z$: the policy *reasons* about its desired affordance and goal to determine its actions.

3. **Post-interaction:** This phase involves separation from the object and any downstream effects on the object, e.g., the purple block coming to rest after releasing it in Fig. 1. For repetitive interactions, this is the same as the next pre-interaction period.

To account for these periods, we detect the robot's *influence* over the environment, e.g., grasping a door handle or lifting a mug, to automatically segment play into the periods above. For detecting single-object interactions, contact signals are readily available with modern robots. The contact signal is smoothed and then chunks of play with contiguous contact are labelled as interaction periods. Several ideas for extending PLATO to handle multi-object interaction detection are discussed in detail in Appendix E. Since play consists of many back-to-back interactions, the post interaction phase of one segment, e.g., retreating from a block and moving towards a cup, is simply the pre-interaction for the next segment, e.g., lifting the cup. We denote pre, interaction, and post windows with superscripts $^{(-)}$, $^{(i)}$, and $^{(+)}$, respectively.

**Sampling Task Relevant Goals Post-Interaction**: By segmenting play into interaction phases, we can more accurately sample an accurate goal environment state. Critically, goals *result* from an interaction by definition. At the end of an interaction, the environment state will very likely have changed. Therefore, rather than arbitrarily labeling the goal as the last state of a 1-2 second window, as in prior work, we can label the goal as any state from the end of the interaction through post-interaction (see $o_g$ in Figure 1). This includes any downstream effects on the object from the interaction (e.g., gravity or inertia). For example, if we slide a block along a table, a valid $o_g$ is any block state before the block stops sliding. With an informative goal sampled, we can proceed to learn a goal-conditioned policy.

**Extracting Affordances from Interaction for Robust Policy Learning**: We define every interaction between the robot and the environment as exploiting some *affordance* on an object in the scene. Affordances are properties of objects that define how they can be used (e.g., a block being grasped, a door knob being turned, or a drawer being opened). Our insight is that learning these affordances (what happens to the object) instead of plans (what happens to the robot) from play will lead to a much simpler and more robust task representation that can operate over varying horizons, and thus will yield much better policies at test time. This paradigm empowers the policy to *reason* about the environment: given access to an affordance (e.g., the door knob being turned) and the goal (e.g, opened door), the policy should be able to work backwards to infer the behavior to exploit that affordance (e.g., reach the knob and rotate the gripper to turn it). This is in contrast to prior work that

relies on randomly selected, short, fixed horizon windows to learn latent representations of *plans*— such plans fail to capture varying horizon tasks and overly depend on the robot state, leading to generalization issues at test time [3].

**PLATO Design**: The PLATO architecture is shown in Fig. 1. At a high level, PLATO learns to model each interaction trajectory $\tau^{(i)}$ as a latent affordance $z$ (**(1) Posterior** in Fig. 1). This latent affordance $z$ *explains* the actions both leading up to (pre-interaction) and during the interaction (see causal structure in Fig. 1). For example, knowing that we want to push a block (goal) and how we want to push it (affordance) allows the policy to infer that it should servo to the correct side of the block first. Therefore, the latent space is learned end-to-end with the policy $\pi$, which decodes latent affordances conditioned on the current state and object goal states to reproduce both pre-interaction and interaction robot actions (**(3) Policy on Interaction** and **(4) Policy on Pre-Interaction** in Fig. 1). By not learning to encode pre-interaction sequences, the critical assumption we make here is that variations in the affordances (interaction phase) are more relevant to the task and thus more important to capture in our latent space than variations across any random behavior sequence (pre-interaction).

The policy action-reconstruction objective alone would encourage $z$ to model robot behaviors (plans), leading to a myopic view of the task. In order to force the latent space to focus on object affordances and thus be more robust at test time, we simultaneously learn a *prior* on the affordance distribution conditioned on just the start and goal object states (**(2) Prior** in Fig. 1), trained to regularize the posterior latent affordance $z$. The posterior on $z$ considers the full window $\tau^{(i)}$, while the prior sees just the start and goal object states, and so the prior helps shape the latent space to encode the affordance in $\tau^{(i)}$ rather than just the action information.

---

**Algorithm 1** PLATO Training

---

1: Given: $H^{(i)}$, $H^{(-)}$, play data $D_{\text{play}}$, interaction criteria $f^{(i)}$,
2: $D_{\text{play}}^{(-)}, D_{\text{play}}^{(i)}, D_{\text{play}}^{(+)} = f^{(i)}(D_{\text{play}})$          ▷ Split into interactions
3: Initialize $E$, $E'$, $\pi$
4: **while** not converged **do**
5:      $\tau^{(-)}, \tau^{(i)}, \tau^{(+)} \sim D_{\text{play}}^{(-)}, D_{\text{play}}^{(i)}, D_{\text{play}}^{(+)}$
6:      Sample $o_g \sim \{o_t^{(+)}\}$
7:      $p(z) \leftarrow E(\tau^{(i)})$          ▷ Posterior Affordance Distribution
8:      $p(z') \leftarrow E'(o_1^{(i)}, o_g)$          ▷ Prior Affordance Distribution
9:      $z \sim p(z)$
10:      $\tilde{a}_{1:H^{(i)}}^{(i)} \leftarrow \pi(s_{1:H^{(i)}}^{(i)}, o_g, z)$          ▷ Policy in Interaction
11:      $\tilde{a}_{1:H^{(-)}}^{(-)} \leftarrow \pi(s_{1:H^{(-)}}^{(-)}, o_g, z)$          ▷ Policy in Pre-Interaction
12:      Compute $\mathcal{L}_{\text{PLATO}}$ with Eq. (1) and update $\pi$, $E$, $E'$.

---

The training procedure is outlined in Alg. 1. After sampling windows from each interaction phase, $\tau^{(-)}$, $\tau^{(i)}$, and $\tau^{(+)}$ (Line 5), as well as a long term goal $o_g$ (Line 6), PLATO encodes the interaction into a posterior and prior affordance distribution (Lines 7-8). Next an affordance $z$ is sampled from the posterior using the reparameterization trick, and $z$ is then used to decode actions during interaction (Line 10) and pre-interaction (Line 11). See Appendix B.1 for more discussion of this training procedure, including its computational efficiency. During training, we utilize action reconstruction losses over both the pre-interaction region and the interaction region windows to train all networks end-to-end. We utilize a KL divergence term to both train the affordance prior network and regularize the affordance posterior network.

$$\mathcal{L}_{\text{PLATO}} = -\log(\pi(a_{1:H}^{(i)}|s_{1:H}^{(i)}, o_g, z)) - \alpha \log(\pi(a_{1:H}^{(-)}|s_{1:H}^{(-)}, o_g, z)) + \beta \, \text{KL}(p(z) \parallel p(z')) \quad (1)$$

Here, $\alpha$ controls the policy focus on reconstructing pre-interaction behaviors, which usually is set to 1. This procedure of learning a posterior and prior network and conditioning the policy on the latent context resembles existing work [3]. The key differences are i) our goals are sampled based on interactions with objects, ii) our latent representations capture object-centric affordances $z$ by intelligently shaping the learning objective, which can lead to more effective and generalizable policies, and iii) the policy reasons over varying and longer horizon sequences by conditioning on the same future desired affordance $z$ and long term goal $o_g$ throughout both pre-interaction and interaction.

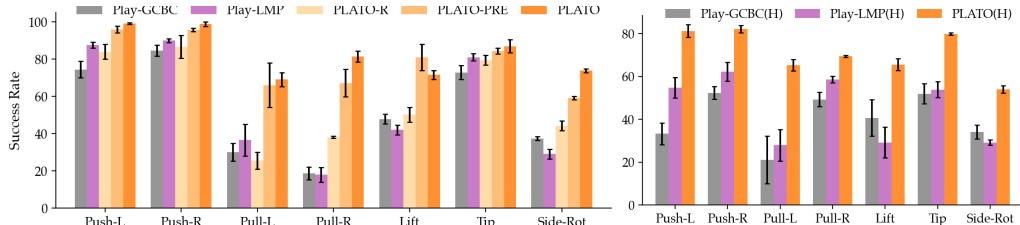

Figure 2: Block2D Success Rates, trained over 3 random seeds and evaluated on various primitives for PLATO and baselines Play-LMP and Play-GCBC. **Left:** Scripted play data, with ablations PLATO-PRE and PLATO-R. **Right:** Human play data. PLATO substantially outperforms baselines on both scripted and human play data.

**Test Time:** At test time, the affordance posterior network $E$ cannot be used to output plans $z$, since it assumes access to a trajectory. Similar to prior work [3], the prior network $E'$ is used to propose $z'$ at test time, using only the current object state $o_t$ and the desired goal object state $(o_g)$. Unlike prior work, the prior network only depends on object states, and will be robust to different robot starting states. Therefore instead of having to predict the exact robot behavior (i.e., plan) to achieve a goal, PLATO predicts an affordance at test time and empowers the policy to exploit this affordance. The policy conditions on $z'$ and goal $o_g$ to produce actions at the current state.

**Addressing Challenges of Prior Work**: By choosing goals $o_g$ resulting from interactions with objects, PLATO better reflects the true demonstrator goal, and thus can reduce the credit assignment problem found with fixed-horizon methods. PLATO handles variable horizon sequences by explicitly training the policy on pre-interaction and interaction sequences together, conditioned on the affordance and long term goal. By biasing the latent space to model *affordances* during interaction, rather than just any random sequence of play, PLATO learns the task relevant behaviors and variations therein to aid in generalization. Our results in Sec. 4 demonstrate the effectiveness of PLATO compared to prior work on a wide range of complex tasks. See Appendix A for further discussion.

## 4 Experiments

In this section, we evaluate our approach extensively across three single-object manipulation environments (**Block2D**, **Block3D-Platforms**, **Mug3D-Platforms**), one multi-object scene (**Playroom3D**), and one real scene (**Block-Real**), across diverse tasks like pushing, lifting, and rotating. These environments, shown in Fig. 3, enable a wide variety of objects and possible affordances. We collect scripted play data in all environments as well as human play data for **Block2D**, and train each method until convergence. See Appendix C for environment, task, data collection, and training details, and Appendix B.2 for method implementation details.

**Baselines:** We compare PLATO against the two state-of-the-art methods for Learning from Play, Play-GCBC (Goal-Conditioned BC) and Play-LMP (Latent Motor Plans) [3]. We also implement two variants of our method, PLATO-PRE, which encodes both the interaction *and* the pre-interaction periods into the latent space (adding $\tau^{(-)}$ as an input to $E$ in step (1) in Fig. 1), and PLATO-R, which replaces the object-centric prior with the prior from Play-LMP (adding the initial robot state as input to $E'$ in Fig. 1). PLATO-PRE results show the sufficiency of affordances as a latent representation, while PLATO-R results show how object-centrism benefits the learned latent space and the policy.

**Block2D Results**: Evaluation results for PLATO for both scripted play data and human play data are shown in Fig. 2, along with results on PLATO-PRE and PLATO-R. On scripted play data, PLATO is able to substantially outperform the baselines on every task. Learning affordances from interactions enables PLATO to model much more complex action sequences, such as those involving the tether action, with high accuracy, even across variations in object dimensions, masses, and initial conditions. On human play data, we again find that PLATO outperforms Play-GCBC and Play-LMP on all of the tasks, emphasizing the ability of our method to scale to human generated data. Interestingly, performance for all methods is worse on the human generated data than scripted data. We attribute this to human data containing many sub-optimal trajectories due to the challenges of teleoperation.

PLATO-PRE, which encodes the pre-interaction period, performs slightly worse but similar to PLATO, validating our hypothesis that interaction trajectories (PLATO) contain sufficient information about the task when compared to also including pre-interaction trajectories in the latent space

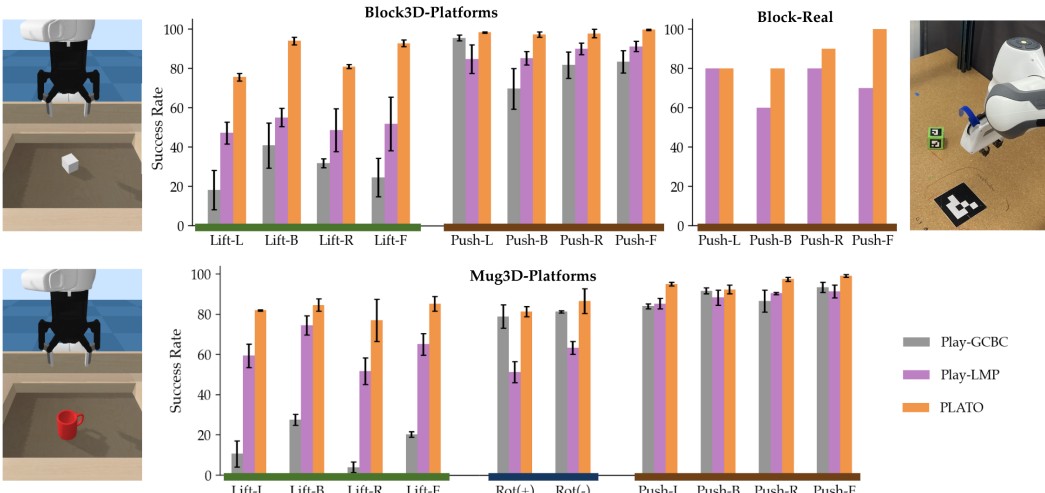

Figure 3: 3D Environment Success Rates, trained over 3 random seeds and evaluated on various primitives for PLATO and baselines Play-LMP and Play-GCBC. **Top Left:** Block3D-Platforms. **Bottom:** Mug3D-Platforms. PLATO substantially outperforms baselines in 3D manipulation environments for pushing, rotating, and lifting tasks. **Top Right:** Block-Real. PLATO trained only in simulation generalizes to real world pushing tasks.

(PLATO-PRE). We find that by adding robot state information to the prior (PLATO-R), performance suffers on most tasks. We hypothesize that this phenomenon is caused by the prior relying too much on the initial state of the robot, and not enough on that of the object. As a result, the latent space will not generalize at test time when the agent inevitably diverges from the offline state distribution. Overall, we see that framing play data through the lens of object interactions (both PLATO and PLATO-PRE) results in much better policy learning than prior state-of-the-art methods.

**Block3D-Platform & Mug3DPlatform Results**: Fig. 3 shows the results of evaluating PLATO on the harder Platform tasks. Interestingly, Both Play-GCBC and Play-LMP do well on the pushing tasks in this setting, but do very poorly on the more complex lifting tasks (and rotate tasks for Mug3D). By modeling an affordance space and properly relating these affordances to goal environment states, our method is capable of recreating diverse types of varying horizon behaviors from play, and unlike Play-LMP, our method scales smoothly as the number of tasks increases.

**Playroom3D Results**: Fig. 4 shows the results on the challenging Playroom3D environment with the drawer, cabinet, and the block, with especially diverse affordances and varying horizon tasks. Once again, PLATO outperforms the baselines on every task. While neither baseline can perform the complex Cabinet-Close primitive, PLATO achieves 100% success. Notably, Play-GCBC performs better than Play-LMP on several tasks, suggesting that the prior network and policy might be out of distribution for the test time states and goals on these complex primitives.

**Block-Real Results**: Fig. 3 shows the results of evaluating PLATO on pushing tasks for a real robot setup with *no additional real world data* (see Appendix C for details). Play-LMP and PLATO both get near perfect success in simulation since these pushing tasks are less complex, but PLATO generalizes better across the gap between real and simulated object dynamics.

Overall, PLATO achieves substantially higher success rates than the baselines on a wide variety of tasks and object properties in simulation and real environments, along with lower variance across random seeds. This trend is even more apparent when we go beyond simple pushing tasks and consider more complex object interactions such as opening and closing doors, drawers, cabinets, lifting, and others. Due to its object-centric view of play, PLATO is able to extract and exploit diverse types of affordances in the environment and can handle varying horizon tasks. See Appendix D for additional experiments including more tasks per scene and a longer discussion of results.

## 5   Conclusion

**Summary**: In this work, we introduced an object-centric paradigm for learning from play data involving segmenting play into a series of object interactions. Prior state-of-the-art methods for

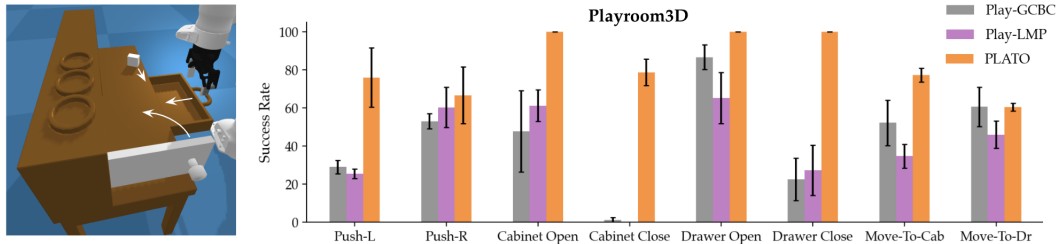

Figure 4: Playroom3D 3D Environment Success Rates. The difference between PLATO and baselines is even greater for more complex and varying horizon tasks, like opening and closing drawers and cabinets. See Appendix D.1 for additional tasks in this environment involving object retrieval and button pressing.

learning from play suffer from credit assignment issues that stem from the short and fixed horizon of sampled trajectories. Our method, PLATO, addresses these credit assignment issues by choosing goals that involve meaningful changes in object state. PLATO learns a latent affordance space to model these interactions and their variations, and simultaneously learns to predict these latent affordances from the goal object state. These latent affordances help to inform the robot behavior across varying horizon tasks. Through our extensive experiments in both 2D, 3D, and real world environments spanning a wide variety of object manipulation tasks, we show that PLATO substantially outperforms prior methods on both scripted and human play data.

**Limitations and Future Work**: Our work introduces a paradigm of learning from variable horizon object interactions, and our method achieves substantially better performance across a variety of *single-object* manipulation tasks. In future work, we intend to expand our notion of interaction to encompass even *multi-object* interactions in play (e.g., tool use). When using tools, the robot might have second and third order effects on the environment that we could model. We believe the interaction paradigm introduced in this work is an important first step to reasoning about these higher order effects. For both the single and multi-object settings, future work might leverage notions of action information density or find bottleneck states to automatically segment interactions.

As shown in Section 4, data collection methods for play greatly affect final policy performance. We primarily evaluate with scripted play, but we hope to collect large human play datasets in future work. Compared to scripted play, we hypothesize that human play consists of much more behavior variability, yielding significant plan and affordance variability for a given start and goal state. Isolating affordances and interactions helps manage this variability, and PLATO is still able to perform well on all the tasks as a result. However, a future direction would be to study how exactly human play data differs from machine generated play data in order to develop more robust methods.

To add, our real robot experiments demonstrate PLATO trained in simulation can generalize to real world dynamics for pushing tasks without any data from the real robot. We hope to collect play data directly on the robot in future work to learn even more complex tasks in the real world. Furthermore, our method makes use of object state information, which may not always be easily available in practice. However, we claim this method can readily handle images with several simple changes to the learned prior, described in Appendix E.

We present thorough discussions of these limitations as well as potential solutions in Appendix E.

## Acknowledgements

This work was funded by JP Morgan, the Office of Naval Research, and NSF Award Numbers 1849952, 1941722, and 2006388.

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
