# OpenReview forum: "PLATO: Predicting Latent Affordances Through Object-Centric Play"
_robot-learning.org/CoRL/2022/Conference — CoRL 2022 Poster_

### Official Review · Reviewer_Qcyf · 2022-08-01

**Originality:** Good
**Technical Quality:** Good
**Clarity Of Presentation:** Fair
**Impact:** 3

**Recommendation:**

Weak Reject: I recommend rejecting the paper, but will not argue for my recommendation if the majority of other reviewers have a different opinion.

**Summary:**

This paper proposes a method to learn latent affordance via object-centric play. The key is to segment the play data sequence into with-contact and with-contact. Given this segmentation, it can generate high-quality training data to learn better latent affordance. Several tasks in both simulation and real-world setup are used to evaluate this method.

**Issues:**

It will be better to demonstrate the proposed method in a complex environment.
Some revisions are required to improve the presentation clarity.

**Quality Of The Limitations Section:**

Additional details required

**Reviewer Expertise:**

3: The reviewer is fairly confident that the evaluation is correct

**Robotics Focus:**

Sufficient demonstration on hardware

**Strengths And Weaknesses:**

Strengths:
- Segmenting the play data is very intuitive, so that better sampling of the initial state and goal state can be achieved for training.
- Several baselines are compared to demonstrate the performance of the proposed method.

Weaknesses:
- All experiments use only one object. It will be much better to evaluate the method on a complex environment with multiple objects, for example, the play data used in play-LMP.
- Segment play data based on contact is intuitive for short-horizon motion requiring only one contact. But if a complex primitive requires contact switching, for example, switching between different tools, this paradigm is hard to handle this kind of long-horizon motions.
- Some paragraphs need move revision to improve the clarity. For example, the paragraph "Learning Affordances during Interaction" in Section 3 is hard to follow.

**Summary Of Recommendation:**

The overall intuition to segment play data based on contact is good. But the experiments' setup is a bit toy. It will be better to demonstrate the proposed method in a complex environment. Some revisions are required to improve the presentation clarity. The proposed method is hard to be applied to complex primitives requiring multiple contacts.

---

> ### Author Response · Authors · 2022-08-20
> **Response to R4**
>
> We thank R4 for their thorough analysis of our approach, experiments, and limitations, and we are excited to work with R4 to improve our paper. We hope to address each of R4’s comments, so please let us know if there are any additional questions or concerns!
>
> ### Complex Tasks
> R4 comments that “the experiments setup is a bit toy” and it is “better to demonstrate the proposed method in a complex environment”. We thank R4 for raising such an important concern, please refer to the “Complex Tasks” section of our shared response above, and please let us know if any additional concerns come up.
>
> ### Multi-Object Scenarios
> R4 states that “All experiments use only one object. It will be much better to evaluate the method on a complex environment with multiple objects.”  R4 further notes that “if a complex primitive requires contact switching, for example, switching between different tools, this paradigm is hard to handle this kind of long-horizon motions.” We thank R4 for raising this key concern, please refer to the “Multi-Object Scenarios” section of our shared response above, and please let us know if you have additional concerns.
>
> In a related comment, R4 notes that one such multiple object environment is “the play data used in play-LMP.” Building on the previous discussion, the environment in Play-LMP is actually multi-object but only single object-interaction. Actually, our Playroom3D environment builds on the same underlying environment assets as Play-LMP, but represents a more complex version of the tasks used in Play-LMP involving *multiple randomized objects* (where objects in Play-LMP were fixed in shape and mass). We remove the simple reaching tasks like the button pressing, but keep the challenging tasks involving object manipulation and articulated object manipulation (drawer opening). Additionally we add a cabinet opening task –  this “door opening” affordance was not included in the Play-LMP prior work, and is significantly harder to learn due to the need to grasp a small handle and perform an arc motion while maintaining the grasp (see Play-LMP and Play-GCBC performance). We believe that the performance of Play-LMP suffers in these settings as compared to prior work precisely because the tasks are more complex for policy learning. We’ve included a longer discussion of this in Appendix C.2, and we plan to release code for our environments and tasks.
>
> ### Other Comments
>
> *“Some paragraphs need more revision to improve the clarity. For example, the paragraph ‘Learning Affordances during Interaction’ in Section 3 is hard to follow.”*
>
> We apologize for the confusion caused here, and we have rewritten much of Section 3, including the “Learning Affordances during Interaction” subsection, to be much more clear and better express our method and key insights.
>
> We hope our response and changes to the manuscript have helped to address R4’s concerns, and in light of this we hope R4 will consider raising their rating. If there are any outstanding questions or concerns, we are happy to discuss further. Thank you!

---

> > ### Author Response · Authors · 2022-08-25
> > **Follow Up Response to R4**
> >
> > We hope our response above has addressed all of your concerns about the paper! We are nearing the end of the rebuttal process, and we really value the chance to discuss and improve our work with you. Before the discussion period ends, please do let us know if you have chosen to raise your rating, and if not, what we can do to address your concerns.

---

> > > ### Comment · Reviewer_Qcyf · 2022-08-26
> > > **Response to authors' comment**
> > >
> > > Thanks for the clarifications and additional experiments. The new room environment and its results look really fantastic. Based on the new info I will update my recommendation to a Weak Accept.

---

### Official Review · Reviewer_7Bgr · 2022-08-01

**Originality:** Good
**Technical Quality:** Very Good
**Clarity Of Presentation:** Excellent
**Impact:** 4

**Recommendation:**

Strong Accept: I recommend accepting the paper and will argue for my recommendation even if other reviewers hold a different opinion.

**Summary:**

- Learn object affordances through unstructured human play, then train a policy to achieve the affordances.
- Key insight: viewing offline play data as diverse object interactions yields better modeling of goals and behaviors. Using automatic segmentation of play into a series of object interactions using proprioceptive information.
- Model: like in [3], learn prior and posterior networks that predict an affordance latent z using KL-divergence. Posterior sees sequence of interaction, while prior only sees start and goal states. The action decoding policy is only trained on the prior latent to ensure that it remains
- Advantages over previous method [3]:
  - dynamic horizon as opposed to fixed short term horizon
  - less credit assignment issues
  - prior more robust to robot starting states
- Envs: block 2d, block 3d, mug 3d, playroom 3d, block real, tasks: pushing, lifting, rotating
- Results: play used as object interactions results in much better policy learning than baselines.


**Issues:**

- “Contact signals are readily available with modern robots. The contact signal is smoothed and then chunks of play with contiguous contact are labelled as interaction periods.”
  - Can you clarify exactly what signal is used (in both real and sim)? torque sensing? pressure sensing (if so specify the limitation that the back of the end effector may not have pressure sensing and cannot be used during play to push things around for example)?
  - if this info is in appendix and I missed it, maybe put a reference to it in the main body?
- “Unlike prior work, the prior network only depends on object states, and will be robust to different robot starting states. Therefore instead of having to predict the exact robot behavior (i.e., plan) to achieve a goal, PLATO predicts an affordance at test time and empowers the policy to exploit this affordance.”
  - is this true from pixels? if not, how do you plan on addressing that when switching to vision? This is an important problem to address.
  - or do you mean that z is more object-focused?
- performance is worse with human data, due to challenges of teleoperation: does that challenge the entire play approach? or having better teleop should fix it?


**Quality Of The Limitations Section:**

Limitations are addressed clearly

**Reviewer Expertise:**

5: The reviewer is absolutely certain that the evaluation is correct and very familiar with the relevant literature

**Robotics Focus:**

Sufficient demonstration on hardware

**Strengths And Weaknesses:**

strengths
- strong results (e.g. much higher accuracy than baselines in playroom) and convincing evaluation
it does well across a variety of tasks, playroom is diverse and hard
- probably the most scalable strategy for diverse data collection method for the real world that exists today, so even though the setups are narrow in the paper, this is a very important problem that is very relevant beyond these narrow setups
- nice implementation details in appendix.
- impressive playroom videos (sim)

weaknesses
- real-world setup not very impressive
- not using image inputs


**Summary Of Recommendation:**

This line of work has potential for major impact because play data might currently be the most scalable strategy for acquiring diverse robotics skills. This paper has some limitations (no vision and limited real setup), but addresses some fundamental problems in earlier learning from play methods and thus brings the field closer to being able to scalably acquire massive sets of diverse skills, which will be a turning point for the field. The model is sound, well explained and yields strong results and robust policies.

---

> ### Author Response · Authors · 2022-08-20
> **Response to R3**
>
> We thank R3 for their “Strong Accept” rating as well as their thorough analysis of our approach, experiments, and limitations, and we are excited to work with R3 to improve our paper. We hope to address each of R3’s comments, so please let us know if there are any additional questions or concerns!
>
> ### Contact Signals in Real/Sim
> R3 asks, “Can you clarify exactly what signal is used (in both real and sim)?” We thank R3 for the comment! The signal used in simulation is the binary contact information between the robot and the rest of the scene, which can easily be computed in pybullet. In the real world, a similar contact sensing process can be done, either with contact pressure sensors mounted on the end effector or with a force/torque sensor on the end effector. Note that contact readings are only used during training (for the purpose of interaction segmentation). Since our real world setup did not involve training on real world data, we did not need to add these additional sensors. However, we believe this modification should be quite straightforward. R3 is correct that with the pressure sensing (which is closest to what we do in simulation), there is a limitation that only interactions with the pressure sensing portion of the end effector will count during segmentation. We have added this discussion to Appendix C.1 in the “Real World Deployment” subsection.
>
> ### Learning from Pixels
> R3 asks, “is this true from pixels? If not, how do you plan on addressing that when switching to vision?” How we should adapt PLATO to learn from images is a very fair question. We thank R3 for raising this key state representation choice. We present several ideas in the “Object State Estimation” shared response above. Please let us know if there are any remaining questions of how to adapt this method to learning from images.
>
> ### Learning from Human Data
> R3 asks if challenges of teleoperation affect the “entire play approach,” or if “having better teleop” will fix this issue. We believe having better teleop will certainly help reduce the mismatch between what a user wanted to do and what they actually did, which would reduce sub-optimality in the play data, and thus improve policy performance. However, this challenge of learning from human data is a more general phenomenon, applying even to imitation learning methods. Imitation learning methods that learn from human data usually require very clean, curated datasets that are not representative of real-world data. We hope to move towards human data with future work. As to why human data is harder to learn from, we speculate that the main issue in learning from human data is in the variability in how a task is demonstrated, which makes policy learning with behavior cloning much more challenging (e.g., multi-modal action distributions, potentially with high variance). This is very much an open research problem that could benefit the robotics community greatly.
>
> We hope our response and changes to the manuscript have helped to address R3’s concerns. If there are any outstanding questions or concerns, we are happy to discuss further. Thank you!

---

### Official Review · Reviewer_KtMz · 2022-08-01

**Originality:** Very Good
**Technical Quality:** Good
**Clarity Of Presentation:** Good
**Impact:** 4

**Recommendation:**

Weak Accept: I recommend accepting the paper, but will not argue for my recommendation if the majority of other reviewers have a different opinion.

**Summary:**

The authors present a novel algorithm to extract manipulation skills from general purpose play interactions. Play produces a stream of unsegmented interaction data that covers a wide range of behaviors and is easy for a human operator to provide. Previous approaches learn a policy from this data by sampling fixed-length short-horizon windows to learn from. However, this can make learning challenging because the intent of the operator is unknown from this short horizon. The insight presented in this paper is that a large portion of manipulation data can be focused around object interaction. The authors show that segmenting play data based on object interactions, and learning latent-policies based instead off these segmented interactions, leads to improved performance when the policies are evaluated on downstream tasks.

**Issues:**

Please see above for more details.
- Dependence on quality of play-data segmentation.
- Limited results on human collected play data.
- Minor clarity concerns/requests for more detail.

**Quality Of The Limitations Section:**

Limitations are addressed clearly

**Reviewer Expertise:**

4: The reviewer is confident but not absolutely certain that the evaluation is correct

**Robotics Focus:**

Sufficient demonstration on hardware

**Strengths And Weaknesses:**

Learning from unstructured play data is a promising way to scale human data collection and the authors do a good job introducing the benefits of this paradigm and the weaknesses of current approaches. Furthermore, the presented insight to naturally segment data from play data using object interactions is intuitive and well presented. The results indeed show that this added structure does help.

There are two main limitations I would like to discuss followed by some minor comments. First, it is unclear how much performance relies on the quality of interaction segmentation. Is perfect interaction segmentation assumed in this work? How robust is the proposed algorithm to segmentation failure or parameters of the segmentation algorithm? (For example, temporarily broken contact in a push action.) I would guess that the scripted data gives cleaner interaction segmentations while segmentation quality in human collected-data may depend heavily on algorithmic parameters.

The second limitation I would like to address is that although play data is motivated as an approach to leverage unsegmentated human-collected data, only the Block2D domain is evaluated using human-trajectories. The more challenging domains all use scripted data which makes both action segmentation easier and doesn’t capture additional difficulties that may arise from more variation in human data. That said, I do appreciate that the performance on human data in the Blocks2D domain outperforms the baselines, even if there is a degradation from scripted data.

 The following are minor comments that I believe will help improve the clarity of the paper allow the reader to better interpret the results:
- How is the human-data collected? Who are the users? What instructions were provided to them?
- I would like to see more discussion about the reliance on the ability to automatically segment play data. How does this limit the proposed approach? Are there tasks that aren’t well modeled this way? What additional infrastructure (software, hardware) is needed for this algorithm to be deployed?
- Can the authors provide some insight into why the proposed method outperforms the Play-GCBC baseline?


**Summary Of Recommendation:**

I recommend that this paper be accepted to CoRL. The authors propose an intuitive way to add structure to “learning from play” algorithms which is inspired by the structure common in manipulation tasks. This approach is a promising research direction for more efficiently using human collected data.

Post-Discussion Update:

I would like to thank the authors for taking the time to respond to my review in detail. I appreciate the additional clarity they provided and the many additional experimental results.

The additional quality of interaction experiments are insightful and do show the algorithm is robust to moderate levels of noise. I would encourage the authors to add motivation for why the fake contacts are only added outside of interactions (in the pre-interaction) as opposed to also breaking contact early during an interaction.

I recommend accepting this paper and encourage the authors to continue investigating the learning from play approaches, especially by including real/unscripted play data for the more complicated tasks.

---

> ### Author Response · Authors · 2022-08-20
> **Response to R2**
>
> We thank R2 for their “Weak Accept” rating as well as their thorough analysis of our approach, experiments, and limitations, and we are excited to work with R2 to improve our paper. We hope to address each of R2’s comments, so please let us know if there are any additional questions or concerns!
>
> ### Quality of Interaction Segmentation
> R2 points out that “it is unclear how much performance relies on the quality of interaction segmentation,” and asks how robust PLATO is to “segmentation failure or parameters of the segmentation algorithm.” We thank R2 for raising this point. To clarify, in all our experiments, we are not assuming access to “perfect” interaction segmentation. In fact, all of our environments will sometimes have imperfect interaction signals due to the demonstrator having notable noise (even in scripted policies, see Appendix C.3 for a discussion of the added noise) – for example, brushing against the table, object, or cabinet door on the way to perform a different task. Intermittent contact is also quite common in all of our play data. However, the smoothing on top of the interaction signal tends to clean many of these signals.
>
> To evaluate the quality of interaction segmentation, we are in the process of conducting an additional ablation experiment in the Block2D environment, where we add artificial noise to the contact labels (e.g., adding “fake contact” regions during pre-interaction and post-interaction segments) to affect segmentation quality. The early results from this experiment show that PLATO is actually quite robust to these fake contact regions, even when these comprise 10% of the interaction data. We will add this new experiment to Appendix C once finished.
>
> Furthermore, we pose this question: what does “perfect” segmentation actually mean? In the framing of our method, any interaction with the environment, even *accidental* ones, are still valid for the affordance space to learn. If we accidentally brush the top of the cabinet door on our way to push an object, and the door opens slightly, this can be seen as a successful cabinet slight-open task. Since the start and goal object state are unique for this task, in theory it should not at all affect the affordance learning for a different start and goal object state. However, accidental interactions will start to affect learning if these interactions are common and bias the policy towards *unsafe* regions of the state (for example, if repeatedly brushing the top of the cabinet door on the way to push the block biases the policy away from pushing the block properly).
>
> We’ve added this discussion to the end of Appendix D.2, and we now show some examples in the data of imperfect segmentation on our website.
>
> ### Use of Human vs. Scripted Data
> We fully agree with R2 that using scripted data “doesn’t capture additional difficulties that may arise from more variation in human data.” One minor point of clarification here is that while play is scripted for several of our environments, as discussed in Appendix C.3, we inject significant unimodal noise of many types into these scripted policies to better resemble human data, for example in trajectory waypoints, speed, and directly in the action space. However, we acknowledge that there is still a gap between scripted and human data as seen in the experimental results for Block2D. Furthermore, imitation learning methods that learn from human data usually require very clean, curated datasets that are not representative of real-world data. We hope to move towards human data in the future, but bridging the gap between clean, curated human or scripted data and raw real-world human data is a much broader, open problem in robotics. This applies not just to play data but also to imitation learning and any other human produced datasets. As R2 notes, we are also encouraged by the fact that PLATO outperforms the baselines on the human play data for the Block2D tasks.
>
> ### Other Comments
>
> *“How is the human-data collected? Who are the users? What instructions were provided to them?”*
>
> We thank R2 for this question, and we have added these details to Appendix C.4, reproduced here for reference. Human play data for the Block2D task is collected using a keyboard control interface. Arrow keys control the ego agent position, while ‘g’ controls the grabbing tether action. One proficient user was used to collect the data, and was shown a set of tasks (the evaluation tasks) to perform during play with the instruction of trying to equally represent each task in their play, but they were also clearly informed that they were not limited to performing just these tasks. This user was given 15 minutes to practice in the environment, after which point data collection began. In future work, we hope to reduce the dependence on balanced, curated datasets and allow play to be truly freeform.

---

> > ### Author Response · Authors · 2022-08-20
> > **Response to R2 (2)**
> >
> > *“What additional infrastructure (software, hardware) is needed for this algorithm to be deployed?”*
> >
> > We thank R2 for this question, as it is imperative to know how we can deploy this system. As described in Appendix C.1 and Figure 7, our initial real robot setup serves as a test setup for real world deployment. It involves multiple cameras mounted at different viewpoints, which give us object state estimates, as well as a known object with ArUco tags. To deploy this system more generically, there are several key additions. In terms of hardware, the robot would need an additional contact sensing patch on its gripper or a force/torque sensor at the end effector to detect interaction with the scene, as well as several cameras to observe the scene. In terms of software, a robust object 6D pose detection algorithm would be utilized to detect the object states under potential occlusion. With these additions, our method should readily scale to many real robot systems. As per R2’s request, we have added these details regarding the real world setup to Appendix C.1.
> >
> > *“Can the authors provide some insight into why the proposed method outperforms the Play-GCBC baseline?”*
> >
> > Appendix A covers many of the challenges in prior work that we tackle with PLATO, and thus lead to the improvement in performance. Play-GCBC suffers from many of the shortcomings listed in this section, specifically the use of short, fixed horizon windows, which can attribute incorrect goals to actions based on the choice in horizon length. In addition, Play-GCBC only attempts to model *what* goals to reach, not *how* we should reach them. The addition of the latent plan in Play-LMP leads to improved performance over Play-GCBC for this reason. Our method retains the latent variable concept but biases the latent space to model affordances rather than just robot plans, which leads to PLATO being able to learn more robust policies, as outlined in more detail in the main text.
> >
> > We hope our response and changes to the manuscript have helped to address R2’s concerns, and in light of this we hope R2 will consider raising their rating. If there are any outstanding questions or concerns, we are happy to discuss further. Thank you!

---

> > > ### Author Response · Authors · 2022-08-26
> > > **Follow Up Response to R2**
> > >
> > > We hope our response above has addressed all of your concerns about the paper! We are nearing the end of the rebuttal process, and we really value the chance to discuss and improve our work with you. Before the discussion period ends, please do let us know if you have chosen to raise your rating, and if not, what we can do to address your concerns.

---

### Official Review · Reviewer_Z8qY · 2022-08-02

**Originality:** Fair
**Technical Quality:** Very Good
**Clarity Of Presentation:** Excellent
**Impact:** 3

**Recommendation:**

Weak Accept: I recommend accepting the paper, but will not argue for my recommendation if the majority of other reviewers have a different opinion.

**Summary:**

This paper proposes a framework for learning goal-conditioned policies from play data. The key insight is to segment the play data based on the robot's contact with objects, instead of splitting the data into fixed-horizon episodes. As a result, each segment of the play data now has a meaningful goal based on the change of object states (referred as "affordance" in the paper), which can be encoded into a latent vector upon which a policy can be conditioned.

**Issues:**

- How are the object states $o$ represented in the network? It should be made clear whether they are explicit states like 6D poses or more like raw RGB(D) observations. These make two very different problems.
- The method segments play into episodes based on contact, and claims that contact detection is easy in modern robots (line 158). However, in multi-object interaction scenarios (such as using tools mentioned in line 146), the environment (object states) can change without the robot directly contacting another object. How to detect the change in goals in these situations?
- Why does it make sense to use latent code from the future for the pre-interaction policy?
- The paper shows that the prior latent code $z'$ is important, but how important is the posterior code $z$? What if the policy network directly conditions on the prior code $z'$?
- In Fig. 2 and 3 there are tasks like Push-L, Push-B, etc. The meaning of "-L" etc. is not explained. I assume it means left, back, right and forward. Why are these considered different tasks and evaluated separately? The performance also varies a lot (80% for Lift-L and 100% for Lift-B in Fig. 3), but they seem the same task to me.

**Quality Of The Limitations Section:**

Additional details required

**Reviewer Expertise:**

4: The reviewer is confident but not absolutely certain that the evaluation is correct

**Robotics Focus:**

Sufficient demonstration on hardware

**Strengths And Weaknesses:**

Strengths
- Very nice presentation. Motivation is clear throughout the paper.
- Simple but effective algorithm design. The meaning of latent codes is much clearer compared to prior work like Play-LMP.

Weaknesses
- Only works on a single-object scene.
- Requires ground-truth object state estimation.
- Not much technical novelty compared to prior work. The contribution is mainly on segmenting the data properly.

**Summary Of Recommendation:**

Although I like the idea of using object states to segment play data, I don't think the contribution of this paper is enough for a good conference paper.

I always think the fixed horizon episodes in play-LMP don't make sense. This paper addresses this issue nicely and demonstrates the benefits of using object-conditioned latent representation. However, it is a pity that the method only works on single-object environments and requires ground-truth object states (e.g. 6D poses), which is very hard to acquire in real environments with clutter. Even the simulation environments themselves are quite simple. The variety of objects is very poor (only blocks and mugs). And it is not clear how different is the test environments from the training ones. The method still has a long way to go from being useful in real-world manipulation.

In addition, I have doubts about the benefits of using play data. It is supposed to contain more diversity and easier to collect. However, in many of the papers that use play data, the play is scripted (meaning diversity is limited) and the environment is fixed. Why not scaling the data collection to more diverse environments and diverse objects? With these simple environments, I don't see the point of having human "play" with it. For most of these simple tasks like pushing and lifting, collecting expert demonstrations per task can be equally or more efficient.

---

> ### Author Response · Authors · 2022-08-20
> **Response to R1**
>
> We thank R1 for their thorough analysis of our approach, experiments, and limitations, and we are excited to work with R1 to improve our paper. We hope to address each of R1’s comments, so please let us know if there are any additional questions or concerns!
>
> ### Novel Contribution beyond Segmentation
> R1 notes that the main novel technical contribution in our work is “segmenting the data properly.” We agree that choosing how to segment the data properly is a major contribution of the work, however we believe that *how* we learn from this data is an equally important contribution. It is not obvious how to adapt the architecture from Play-LMP to work with properly segmented trajectories (e.g., what segments to encode, what portions of state space should each model see, which segments the policy can be trained on, etc.), or even what proper segmentation is and why it should help. See Section 3 for these method details and ablation experiments for further analysis. Additionally, we believe our more general contribution is an object-centric *paradigm* for learning from play, which prioritizes modeling variations in object trajectories (i.e., affordances) rather than robot behaviors, and then learns policies to exploit these affordances across varying length sequences. In essence, we are structuring policy learning to *link* the correct behavior to a desired object affordance without significant prior environment knowledge, and we then can infer the correct behavior at test time. This process of reasoning over multiple skills stands in stark contrast to prior methods.
>
> In addition, while there are many technically novel elements to our approach for learning, we agree with R1 that the simplicity of our method and key insight is a strength rather than a limitation: simple methods are often the most important for creating robust robotic systems, as they are often easier to implement and more likely to have widespread adoption. Setting aside the technical novelty of this approach, we believe the substantial gap in performance between PLATO and prior work demonstrates the importance of these contributions.
>
> In order to clarify this contribution and the details of our method, and to address concerns of both R1 and R4, we have reorganized and rewritten much of Section 3.
>
> ### Complex Tasks
> R1 comments that we should be “scaling the data collection to more diverse environments and diverse objects,” and that our environments are too simple to benefit from play. We thank R1 for raising such an important concern, please refer to the “Complex Tasks” section of our shared response above, and please let us know if you have additional concerns.
>
> ### Benefits of Using Play Data
> R1 questions the “benefits of using play data” in settings where “play is scripted and the environment is fixed”. We completely agree with R1 that as discussed in prior work, play data is helpful for learning since it contains more “diversity” than demonstration data. However, this diversity is not only in *how* behaviors are performed (e.g. is the policy scripted or human), but also in the chaining of different behaviors back to back, which lands us in new initial states for the next task. In other words, play allows us to see more of the state space by expanding the initial state distribution for all tasks *without any designer effort*. Now in theory one can design the same broad initial state distribution for all tasks manually, but this involves significant effort and planning both in specifying tasks and performing the resets manually. Play gets you all this state coverage for free. While not the ideal source of data, scripted play still captures this type of goal diversity.
>
> Another point of clarification here is that while play is scripted for several of our environments, as discussed in Appendix C.3, we inject significant unimodal noise of many types into these scripted policies to better resemble human data, for example in trajectory waypoints, speed, and directly in the action space. However, we acknowledge that there is still a gap between scripted and human data as seen in the experimental results for Block2D. Furthermore, imitation learning methods that learn from human data usually require very clean, curated datasets that are not representative of real-world data. We hope to move towards human data in the future, but bridging the gap between clean, curated human or scripted data and raw real-world human data is a much broader, open problem in robotics.
>
> ### Multi-Object Scenarios
> R1 mentions that our method only works on a “single-object scene”, and that “in multi-object interaction scenarios (such as using tools mentioned in line 146), the environment (object states) can change without the robot directly contacting another object.” We thank R1 for raising this key concern, please refer to the “Multi-Object Scenarios” section of our shared response above, and please let us know if you have additional concerns.

---

> > ### Author Response · Authors · 2022-08-20
> > **Response to R1 (2)**
> >
> > ### Object State Estimation
> > R1 correctly points out that our presented approach relies on “ground-truth object state estimation.” We thank R1 for raising this key state representation choice, please refer to the “Object State Estimation” section of our shared response above, and please let us know if you have additional concerns.
> >
> > ### Other Comments
> >
> > *“How are the object states o represented in the network?”*
> >
> > Object states are 6D poses similar to prior work as noted in the Real Robot Environment details section of Appendix C.1, and we have now also included these details when discussing data collection Appendix C.4 for further clarification.
> >
> > *“Why does it make sense to use the latent code from the future for the pre-interaction policy?”*
> >
> > Based on both this comment and R4’s concern, we decided to rewrite the Learning Affordances during Interaction part of Section 3 to be more clear and better express our method. We apologize for the confusion caused here. To answer your question, there are several reasons for conditioning on the future latent code. Firstly, this allows us to use the same policy for all parts of evaluation (we don’t need two separate policies pre and during interaction). Since the prior network only sees object states, during all of pre-interaction the predicted affordance distribution will be the same as that at the start of interaction, and so the input to the pre-interaction policy will be consistent with what we see during training (this is what we mean in the text when we say this puts the policy in-distribution at test time). Secondly, more intuitively latent code represents *how* we want to move an object (e.g., moving a block forward 0.5m quickly), and the robot should know what we want to do in order to position itself to accomplish this (e.g, servo to the closer side of the block with some margin depending on desired speed). The goal alone (e.g., the final position of the block), may not contain all the information necessary to learn a good pre-interaction policy. A natural question might be why not give the latent encoder full knowledge of both the pre-interaction states and the interaction states to help policy reconstruction? We experiment with this ablation in PLATO-PRE, and show that it does not improve performance as compared to PLATO, suggesting that as claimed in the paper, the pre-interaction policy doesn’t benefit from information about the whole pre-interaction trajectory, and the future affordance is enough to train a good pre-interaction policy.
> >
> > *“The paper shows that the prior latent code z′ is important, but how important is the posterior code z? What if the policy network directly conditions on the prior code z′?”*
> >
> > Interestingly, removing the posterior and conditioning on the prior code z’ during training reduces to performing Play-GCBC with a slightly modified architecture. This is because the policy sees a superset of the inputs that the prior sees, so the prior just becomes an unnecessary part of the policy. If you keep the posterior and keep the KL penalty, but still condition on the prior plan z’ during training, the posterior will now not be used at all for learning since it can trivially fit the prior plan distribution z’ just by ignoring all the intermediate states of the trajectory, leading to a zero KL penalty. Therefore, this also reduces to Play-GCBC.
> >
> > *“Why are [Push-L and Push-R] considered different tasks and evaluated separately?”*
> >
> > Each pushing task is really push <direction> + sizeable goal variation (i.e. distance, speed, etc). So the distinction was chosen to separate by direction to more accurately reflect tasks that we viewed as having similar *affordances*. Figure 8 in the Appendix provides some intuition that our method might separate the affordances by direction, and we want to study how our method learns different *affordances* rather than *tasks*. We now reflect on this evaluation choice in Appendix C.3.
> >
> > *“The performance also varies a lot (80% for Lift-L and 100% for Lift-B in Fig. 3), but they seem the same task to me.”*
> >
> > One hypothesis is that there could be some biases in the data since this seems to be common across methods, and interestingly the performance tends to be closer for directions along the same axis (e.g. for left and right vs. forward and back). This also highlights why we separate task evaluation by direction (e.g., push left) rather than only by the semantic task label (e.g., push). This discussion has been added to Appendix D.2.
> >
> > We hope our response and changes to the manuscript have helped to address R1’s concerns, and in light of this we hope R1 will consider raising their rating. If there are any outstanding questions or concerns, we are happy to discuss further. Thank you!

---

> > > ### Author Response · Authors · 2022-08-25
> > > **Follow Up Response to R1**
> > >
> > > We hope our response above has addressed all of your concerns about the paper! We are nearing the end of the rebuttal process, and we really value the chance to discuss and improve our work with you. Before the discussion period ends, please do let us know if you have chosen to raise your rating, and if not, what we can do to address your concerns.

---

### Author Response · Authors · 2022-08-20
**Shared Response**

We thank all the reviewers and the meta-reviewer for their detailed reviews and valuable insights. First we would like to respond to some general concerns that multiple reviewers raised. Under each review, we will provide additional responses to each reviewer’s specific concerns.

### Complex Tasks (R1, R4, Meta Reviewer)

Several reviewers noted that they would like to see more diverse tasks and objects. We agree that it is critical to evaluate our method on diverse tasks and objects. However, we believe we have done a significant amount of evaluation on diverse tasks and objects and we respectfully disagree that the evaluated tasks are too simple and that the objects are not diverse.

First, we have incorporated diversity in primitives and variations in objects, which we emphasize is crucial and is not present in prior work such as Play-LMP. The tasks themselves cover a number of diverse primitives, where each “primitive” consists of variations in the goal, for example varying pushing distance or lifting placement location. The motions of the scripted primitives also have sizable variations in intermediate waypoints, speed, etc. Within *all of our environments*, we inject large variations in the positions, orientations, each size dimension, and masses of each of the blocks and mugs, which each require different strategies from the robot’s perspective. Furthermore, grasping the mug involves a very precise interaction with the handle, which contrasts the wide, centered grasp used with blocks. This is in comparison to the closest prior work, i.e., Play-LMP tasks, which usually are projected to far fewer primitives and variations in the policy.  The Play-LMP tasks largely involve either fixed object shapes with limited random pose initialization or static scene elements like buttons and constrained unchanging drawers. This might make it seem that these tasks are complex at the surface visual level, but we argue that our set of tasks and primitives require a much greater *range of behaviors*, such as grasping different shapes, rotating blocks to a wide spectrum of new orientations, and handling a variety of block masses, and thus these tasks are more complex. We would like to emphasize that visually interesting environments (e.g., added buttons or static objects as in Play-LMP), do not really add to the complexity of policy learning. What makes policy learning challenging is variations in behaviors and object properties, which we extensively test with our experiments. We believe that the performance of Play-LMP suffers in these settings precisely because the tasks are more complex for policy learning.

Second, our experiments include a visually interesting and complex environment, Playroom3D, which represents a more challenging version of the tasks used in Play-LMP involving some of the same underlying assets but having *multiple randomized objects*. To make the tasks even more challenging, we added the cabinet door opening and closing tasks. This “door opening” affordance was not included in the Play-LMP prior work, and is significantly harder to learn (see Play-LMP and Play-GCBC performance). We thus believe that our tasks are significantly more complex and diverse compared to prior work in this domain and adequately demonstrate the performance of our algorithm and a significant gap with prior work. We’ve included this more thorough discussion in Appendix C.2, and we plan to release code for our environments and tasks. We also point the reviewers to the generalization experiment in Appendix D.1 and Table 5 that demonstrates the robustness of PLATO to state/action/goal distribution shift at test time, which can be common in real world scenarios.

This being said, we are open to the reviewers’ suggestions during this rebuttal period for any other tasks that would help to ease their concerns about task and object diversity while still representing a reasonable advancement from prior work. We have additionally evaluated two more challenging tasks in the Playroom3D environment that were demonstrated during play but not used for our original evaluation: From-Cabinet and From-Drawer (pull object out of open cabinet, lift it out of open drawer), again with substantial object and primitive variation as discussed above. The performance on these tasks are lower compared to the other tasks as they require many pre-conditions and thus are not equally represented during our collected play data. However, even with fewer examples in a crowded dataset of other tasks, there is still a large gap between PLATO and the next best method.

|                 | From-Cabinet      | From-Drawer |
| ----------- | ----------- | ----------- |
| Play-GCBC | 6.3 $\pm$ 4.1   | 3.7 $\pm$ 0.33       |
| Play-LMP    | 1.0 $\pm$1.0   | 19.7 $\pm$ 6.5        |
| PLATO         | 11.7 $\pm$ 1.8   | 58.3 $\pm$ 3.5        |

---

> ### Author Response · Authors · 2022-08-20
> **Shared Response (2)**
>
> From-Cabinet requires a novel end effector orientation and grasping procedure in order to avoid collision with the cabinet and table walls. We speculate that due to the novel motion, limited examples and the presence of other tasks in the data, all methods do notably worse on the From-Cabinet task, however From-Drawer performance for Play-LMP and PLATO is closer to To-Drawer performance since these tasks involve similar object lifting behaviors.
>
> We would also like to emphasize that in all of our tasks PLATO shows a significant gap compared to the baselines, suggesting that these tasks are past the frontier of what learning from play methods could previously accomplish. Furthermore, the policy performance of the baselines varies tremendously between our spectrum of tasks, being quite low for more complex tasks (e.g., the cabinet tasks in the Playroom3D), but higher for the easier tasks (e.g., the pushing tasks). This variation in policy performance is evidence that our tasks have good coverage over different levels of task complexity.  Our goal in this work is to illustrate the potential of PLATO to handle a wide range of complex affordances, and we think there is sufficient evidence from these results for practitioners to try out our method in their respective domains.
>
> We’ve added a more thorough discussion to Appendices C.2 and D, and we have incorporated this new table as part of Table 6. In light of these changes we hope the reviewers agree that our tasks sufficiently demonstrate that PLATO achieves much more than an incremental improvement over prior work.
>
> ### Multi-Object Scenarios (R1, R4)
> As we mention in the limitations section, we fully agree that multi-object interaction scenarios like tool-use represent a key challenge for future work. However, we’d like to point out that there is a distinction between a single object *scene* and a single object *interaction*. For example, Playroom3D involves several objects in the scene at a given time (object, drawer, cabinet), however the multi-object interactions are limited to the object being placed on the drawer, the cabinet, or the table (i.e., dynamic object interacting with a semi-static one). We posit that many (if not most) commonly studied robotics environments fall under this multi-object scene, single-object interaction paradigm, and so we believe our method as presented would be quite applicable to many realistic robotics domains of interest.
>
> On a separate note, we agree that scaling to multiple dynamic object interactions (e.g., tool use) is critical in future work, but we believe that our method is amenable to multi-object settings with some modification. We will give a couple of ideas of how future work might tackle this problem in the hopes of opening up a broader discussion. Before these ideas, one general point of clarification: PLATO introduces detecting “interaction” as a more general concept, which applies even when no contact occurs between the robot and the desired object (e.g., tool use). We used contact as our interaction signal since it was the most readily available for single-object interactions. However for multi-object interactions, the notion of interaction still exists and our method still applies if we can detect these interactions somehow. Since PLATO only requires detecting interaction during training, one option is to have a human label interaction segments in their training data manually. If this is too time intensive, we can potentially leverage *learned* binary signals for interaction using limited supervised interaction labels. Consider the tool-use task of hitting a hockey puck with a hockey stick. Even though we cannot directly observe a contact signal between the stick and the puck, future work could *learn* to predict the interaction signal from visual (e.g. observing the stick hit the puck) and maybe even haptic information (force feedback of the hockey stick on the end-effector when hitting), using supervised labels. In addition, recent approaches like ComPILE [1] have learned to segment skills without any labels, and could be used to isolate an interaction signal for dynamic multi-object interactions in a self-supervised fashion.
>
> We also are in the process of conducting a new ablation experiment to determine the effect of poor quality interaction segments, and early results there show that PLATO is actually quite robust to imperfect segmentation. This also supports the ideas above of learning to detect multi-object interaction, in case the learned signal is imperfect. While designing such a system for detecting interaction might require some effort, we believe that PLATO’s results suggest that such effort can result in serious performance gains for policy learning from play. We have clarified our wording to reflect this general “interaction” paradigm in the main text in Section 3.

---

> > ### Author Response · Authors · 2022-08-20
> > **Shared Response (3)**
> >
> > With this general notion of interaction in mind, we would like to present two potential ideas for future work:
> > 1. One idea would be to learn an *embodiment-specific* affordance space. Then, each tool can be viewed as a different embodiment of the robot, and with knowledge of what tool is currently being used, we can learn affordances specific to that tool. At test time if we know what tool we picked up, the policy can leverage the affordance space of this particular tool in a manner similar to PLATO.
> > 2. As another related idea, we could attempt to learn multiple *degrees* of interaction: e.g. similar to how PLATO learns the relationship between an object affordance (hockey stick moves) and robot skill (grasp and swing a stick), we might also learn the relationship between a second order object affordance (hockey puck moves) and a first order object affordance (hockey stick strikes). Then at test time the policy could reason backwards in time, first inferring the correct second order affordance (hockey puck moves), then the first order affordance (hockey stick swings), then the robot action to take (grasp and swing the stick).
> >
> > We see our work as a first step towards exploring some of these interesting paradigms for interaction and multi-level skill reasoning. As an important caveat, both our work and prior work are *not* tackling the long horizon learning problem, where the robot is evaluated at chaining multiple tasks together. Planning for tool-use is inherently a long horizon task, involving separate phases of first acquiring the tool and then using it, then putting it down. While out of scope for both this work and prior work, we hope to tackle this in future work using the ideas discussed here.
> >
> > We have added this discussion to Appendix E, and in light of this we hope the reviewers agree on both the applicability of PLATO to many realistic real-world tasks, as well as the potential of PLATO to be applied to multi-object dynamic interactions.
> >
> > ### Object State Estimation (R1, R3)
> > Several reviewers questioned our use of object state estimation rather than learning from images. Firstly, there is a sizable research field devoted to improving object state estimation using learning and filtering techniques (even involving state estimation under clutter), so we believe that object state estimation methods will get even more practical in the coming years. Secondly, as we show in the PLATO-R ablation, isolating object state information in the prior actually *helps* policy learning, suggesting that adding in ground truth object state estimation might justify the implementation effort. Finally, if these hypotheses about object state estimation do not hold, while we did not explore learning from images, we believe our method can readily scale to images with a few modifications. Critically, the only obstacle to extending our method to image inputs is our learned prior network, which utilizes just ground-truth object state information rather than the full proprioceptive state and object state. Note that our PLATO-R ablation can be extended to learn from images without any additional modifications, however this ablated method lacks the benefits of object-focused affordance learning (see Section 4). One method would be to mask out only the object(s) of interest from the start and goal images before passing them into the prior to encourage a robot agnostic latent space. Another method would be to learn an object representation directly from images that is independent of the robot state (e.g. with contrastive learning on negative examples of different robot poses, but identical environment states). Regardless, we argue that the results in this paper show that learning object centric representations actually improves the robustness of policies at test time, and thus justifies the extra effort of designing these representations. We have added this point to the Limitations section as well as this more thorough discussion in Appendix E.
> >
> >
> > [1] Kipf, Thomas, et al. "Compile: Compositional imitation learning and execution." International Conference on Machine Learning. PMLR, 2019.

---

> > > ### Author Response · Authors · 2022-08-20
> > > **Updated PDF**
> > >
> > > **Comment:**
> > >
> > > Revised PDF + Supplemental Based on Reviewer Comments.
> > >
> > > **Zip File:**
> > >
> > > /attachment/9b1aebf63549b6d9caecc539907ab1fbd863d563.zip

---

> > > > ### Author Response · Authors · 2022-08-26
> > > > **UPDATE: New Experiments**
> > > >
> > > > We have run several additional experiments to address concerns raised by the reviewers. We first report the full results from the quality of interaction ablation experiment. In addition to the From-Drawer and From-Cabinet experiments discussed previously, we add to the number of tasks and scene objects with new button pressing tasks in Playroom3D.
> > > >
> > > > ### Ablation of Quality of Interaction Data
> > > > We run an additional ablation experiment in Block2D to demonstrate the robustness of our method to ``false” interaction signals. Here we artificially add fake contact signals outside of interactions (e.g., during pre-interaction), causing false interactions to be segmented during training. We evaluate PLATO-FC(%), where % denotes the percentage of *interactions* that are false positives. We show results for a single seed of each PLATO for 4%, 8%, and 12%.
> > > >
> > > > |                 | Push-L    | Push-R | Pull-L | Pull-R | Lift | Tip | Side-Rot |
> > > > | ----------- | ----------- | ----------- |----------- |----------- |----------- |----------- |----------- |
> > > > | PLATO | 99.1 | 98.8 | 69.0 | 81.4 | 71.5 | 86.9 | 73.8 |
> > > > | PLATO-FC(4%) | 100 | 95.9 | 61.3 | 57.0 | 78.4 | 93.0 | 68.3 |
> > > > | PLATO-FC(8%)  | 97 | 96.7 | 35 | 65.6 | 80.4 | 95.6 | 48.7 |
> > > > | PLATO-FC(12%)  | 94.8 | 100 | 61.5 | 10.1 | 60.8 | 92.3 | 63.6 |
> > > >
> > > > Pulling tasks have some variance in performance under added false contact, however considering how much data is affected, PLATO is quite robust to false contact signals for all tasks.
> > > >
> > > > ### New Tasks for Playroom3D
> > > > To further address the reviewers’ concerns about object / task variety and complexity, we have added a new set of tasks to the Playroom3D environment which more closely align with the tasks in Play-LMP. These tasks involve interacting with buttons in the cabinet, which are analogous to reaching tasks, but involve a pressing motion on a relatively small button. Similar to Play-LMP, these buttons are static, however, unlike Play-LMP the button reaching is partially obstructed by the cabinet door. This brings the total number of tasks to 12 (not including goal randomization) in the Playroom environment across 5 scene objects. We believe this is an adequately complex environment to evaluate our method. Performance for other tasks when trained with the new button tasks is similar to the reported results, and thus left out here for conciseness.
> > > >
> > > > |                 | Button 1     | Button 2|
> > > > | ----------- | ----------- | ----------- |
> > > > | Play-GCBC | 68.7 $\pm$ 10.7   | 66.7 $\pm$ 17.5       |
> > > > | Play-LMP    | 63.3  $\pm$9.2   | 42.3 $\pm$ 13.2        |
> > > > | PLATO         | 79 $\pm$ 2.1   | 100 $\pm$ 0       |
> > > >
> > > > As discussed in the first response, these tasks are quite simple to solve, in comparison to our initial set of evaluation tasks. However, PLATO still performs substantially better and with more consistency on these new button tasks.
> > > >
> > > > In light of these new experimental details, we hope the reviewers’ concerns have been addressed and will consider raising their rating.

---

### Meta-Review · Area_Chair_cRyP · 2022-08-15

**Recommendation:** Accept (Poster)
**Confidence:** 4

**Metareview:**

The paper provides an interesting concept of using object-centric play data for learning multi-task policies with a sound framework. The paper is well presented and the reviewer concerns were well addressed including improving the experiments section. Including more complex tasks remains a concern and it can significantly improve the paper, if addressed. However, with the current experiments, the paper makes sufficient contributions to the community.

**Best Paper Nomination:**

No

---

> ### Author Response · Authors · 2022-08-20
> **Response to Meta Reviewer**
>
> We thank the meta-reviewer and all the reviewers for their comments and critiques. We are excited the reviewers agree that our proposed approach is both simple and effective, that PLATO tackles the important issue of learning from varying horizon tasks in play data, and that the results of PLATO demonstrate substantial improvement over prior work.
>
> For ease of review, we summarize our responses to the major critiques of PLATO below. More detail can be found in the shared responses below and/or in the individual reviewer responses.
>
> ### Tasks Are Not Complex
> We add significant randomization in object properties to our environments, and our tasks cover a wide range of affordances involving free objects and articulated ones, as well as a wide range of robot behaviors. As evidenced by the low performance of baselines and the significantly higher performance of our method, these tasks are actually *more complex* than the tasks used in prior work, and represent an advancement of the frontier of what we can reasonably learn from play data. In order to add to the tasks we consider and help to illustrate this point, we run additional experiments for two new tasks in the Playroom environment. Additionally, we add button pressing tasks similar to those in the Play-LMP paper to the Playroom environment.
>
> ### Handling Multi-Object Scenarios
> While this is a limitation of PLATO, as we noted in the Limitations section, we believe future work can adapt PLATO to multi-object scenarios with only few modifications. We have outlined several ideas for how this can be done, and we added a more thorough discussion of the limitations of automatic interaction segmentation.
>
> ### Use of Scripted Play Data
> While many experiments utilize scripted play data in place of human play data in this work, we inject significant amounts of noise in many forms into the scripted policies to help bridge the gap between scripted and human data. As evidenced by the Block2D human data results, we acknowledge this does not completely remove the gap, and our future work should aim to better understand the differences between human and scripted data. However, in practice most imitation learning methods rely on clean, curated datasets. Learning from raw, real-world human data is a more broad problem for the research community to address. We clearly state these limitations in the Limitations section and the Appendix.
>
> ### Use of Object State Estimates instead of Images
> Only the learned prior in PLATO only sees object states in our formulation. There is a sizable research field in learning to estimate object poses from images, and we believe these methods will become even more practical in the coming years. While our work does not attempt to learn from images, we additionally outline several ideas for how PLATO’s learned prior can be trained solely with images. Furthermore, our ablations in the main text demonstrate that having object specific representations is well worth the added cost of designing such representations.
>
> ### Specific Manuscript Changes
> We also provide a list of specific changes to the manuscript and additional experiments based on the reviewers’ comments. Changes in our manuscript are shown in blue.
> - Experimented with two new tasks in the Playroom3D environment, discussed in greater detail in Appendix D.1 and added to Table 6
> - Added additional button tasks in the Playroom3D environment.
> - New Ablation experiment for the quality of interaction segments
> - Significant rewrite of Section 3 to more clearly explain PLATO and its assumptions along with Fig. 1 change.
> - Addition of more training procedural details to Appendix B.1
> - Real world deployment requirements of PLATO in Appendix C.1
> - Discussion of task complexity in Appendix C.2
> - Clarified choice of evaluation tasks in Appendix C.3
> - Added user information and human play collection details to Appendix C.4
> - Discussion of Interaction Segmentation Quality in the end of Appendix D.2
> - Discussion of limitations (multi-object scenarios, object state estimation) in Appendix E and Section 5.